# Embedding Play to Enrich Physical Therapy

**DOI:** 10.3390/bs13060440

**Published:** 2023-05-24

**Authors:** Alyssa LaForme Fiss, Ragnhild Barclay Håkstad, Julia Looper, Silvana Alves Pereira, Barbara Sargent, Jessica Silveira, Sandra Willett, Stacey C. Dusing

**Affiliations:** 1School of Physical Therapy, Texas Woman’s University, Dallas, TX 75235, USA; afiss@twu.edu; 2Department of Health and Care Sciences, Faculty of Health Sciences, UiT The Arctic University of Norway, 9037 Tromsoe, Norway; 3School of Physical Therapy, University of Puget Sound, Tacoma, WA 98416, USA; jlooper@pugetsound.edu; 4Department of Physiotherapy, Universidade Federal do Rio Grande do Norte, Natal 59078970, Brazil; silvana.alves@ufrn.br; 5Division of Biokinesiology and Physical Therapy, Herman Ostrow School of Dentistry, University of Southern California, Los Angeles, CA 90033, USA; bsargent@pt.usc.edu (B.S.); stacey.dusing@pt.usc.edu (S.C.D.); 6Department of Physical Therapy, Texas State University, Round Rock, TX 78665, USA; jm2049@txstate.edu; 7Munroe-Meyer Institute, University of Nebraska Medical Center, Omaha, NE 68198, USA; swillett@unmc.edu

**Keywords:** play, physical therapy, infants, toddlers

## Abstract

Play is an active process by which an individual is intrinsically motivated to explore the self, the environment, and/or interactions with another person. For infants and toddlers, engaging in play is essential to support development across multiple domains. Infants and toddlers with or at risk of motor delays may demonstrate differences in play or challenges with engaging in play activities compared to typically developing peers. Pediatric physical therapists often use play as a modality to engage children in therapeutic assessment and interventions. Careful consideration of the design and use of physical therapy that embeds play is needed. Following a 3-day consensus conference and review of the literature, we propose physical therapy that embeds play should consider three components; the child, the environment, and the family. First, engage the child by respecting the child’s behavioral state and following the child’s lead during play, respect the child’s autonomous play initiatives and engagements, use activities across developmental domains, and adapt to the individual child’s needs. Second, structure the environment including the toy selection to support using independent movements as a means to engage in play. Allow the child to initiate and sustain play activities. Third, engage families in play by respecting individual family cultures related to play, while also providing information on the value of play as a tool for learning. Partner with families to design an individualized physical therapy routine that scaffolds or advances play using newly emerging motor skills.

## 1. Introduction

Children, defined here as those under the age of 3, rely on parents or caregivers to provide a safe, dependable, and calm environment in which to live, grow, and learn. This environment is impacted by the child’s ability to self-direct their movements, engage with objects, and lead interaction with their caregivers. This often takes on the shape of play. In this paper, we define play as an active process by which an individual is intrinsically motivated to explore the self, the environment, and/or interactions with another person. It is enjoyable with a natural flow individually or between participants. Play is valued for its own sake; the means are more valuable than the ends. This definition was developed during a 3-day consensus conference, organized and hosted by the Motor Development Lab at the University of Southern California.

Multiple organizations have highlighted the importance of play in the developmental continuum. The United Nations Convention on the Rights of the Child recognizes the right “to engage in play and recreational activities appropriate to the age of the child” as a fundamental right for every child [1] (Article 31). Likewise UNICEF highlights the importance of “playing to learn” as an important piece of early childhood education and childcare [2]. The American Academy of Pediatrics (AAP) suggests pediatricians concerned about a child’s development write the family a prescription to play with their child, as play may be as powerful as any medication [3].

## 2. Importance of Play in Early Childhood

Play is essential to support the development of multiple developmental domains: motor, language, cognition, social-emotional, and adaptive behavior [3,4,5]. Play allows children to learn about themselves and their environment [3,5,6]. Early reciprocal caregiver interactions such as eye contact, smiling, and mimicking sounds are some of the earliest forms of play, laying a foundation for future socialization and language development [7]. As the child ages, exploring the environment during play provides opportunities for children to learn what their body can do and to practice skills that support the development of new abilities [8,9]. For example, an infant may see an interesting toy out of reach, and through repeated attempts to obtain the toy, develop new motor skills such as rolling or crawling. As motor skills advance, new opportunities for exploration further facilitate cognitive growth.

### 2.1. Play in Pediatric Physical Therapy

Pediatric physical therapists, who traditionally serve children and youth from birth to 21 years of age, and early childhood service providers use play as a modality to assess development and to engage children in intervention [5,10]. Physical therapists design play involving activities representative of the child’s developmental stage and gradually adapt the environment to introduce novelty and challenge [11]. Active play during therapeutic intervention in the first 3 years of life is crucial to maximize participation and function, affect positive neuroplasticity, and promote the development of a sense of self. The intentional use of play increases motivation, a critical modulator of neuroplasticity and engagement in physical therapy [12,13]. Recent research on interventions to support early development, including Supporting Play Exploration and Early Developmental Intervention (SPEEDI) [13] and Goal Activity Motor Enrichment (GAME) [14] studies, incorporate principles of play as important components of the intervention, yet define play differently, highlighting the importance and need for additional consideration of play. Additionally, when Håkstad and colleagues [11] observed pediatric physical therapy (PT) sessions with a focus on play, they noted that the therapists occasionally interrupted or prematurely redirected infant play focusing instead on specific therapeutic goals or therapeutic handling. Similarly, in a study comparing two PT intervention approaches, variations on the amount of “help” provided to the child and how the toys were used clearly distinguished the approaches [15]. Because of the importance of play, we encourage thoughtful reflection on the use of play in pediatric PT to ensure the therapist is aware of and sensitive to the child’s cues and responses to facilitate ongoing, interactive play.

### 2.2. Play of Infants and Toddlers with or at Risk of Motor Delays

There is a relative paucity of research on the play of children with or at risk of motor delays to inform PT strategies and family coaching. This lack of research is concerning since young children with motor delays require a supportive environment to fully engage in play. This environment may include adaptive toys, adaptive equipment, and high levels of parent and caregiver responsiveness and support [16,17]. Family coaching on ways to support play is critical since parent and caregiver involvement in a child’s play can enhance the complexity, duration, and frequency of more advanced play behaviors [18].

Due to their underlying medical or developmental condition, children with motor delays may demonstrate differences in play relative to their peers with typical development. For example, infants with autism spectrum disorder demonstrate differences in exploratory play, including atypical use of objects for sensory stimulation, more repetitive interactions with objects, and prolonged visual inspection of objects from odd angles [19,20,21]. Toddlers with autism spectrum disorder are also less likely to engage in symbolic play [22]. As infants, children with Down syndrome demonstrate differences in exploratory play that are associated with a lack of object mastery and decreased attention to objects [18], and as toddlers they tend to repeat the same play schemes more often than their peers with typical development [23]. For young children with cerebral palsy, higher playfulness has been associated with higher gross motor function, more effective adaptive behavior, and less impact of their health conditions on daily life [24]. An understanding of the impact of a child’s condition on their play is an important consideration when designing intervention programs and family coaching to meet therapeutic goals.

## 3. Embedding Play to Enrich and Individualize Physical Therapy

### 3.1. Engaging the Child

#### 3.1.1. Respecting the Child’s Behavioral State and Cues

Engaging the child in PT that embeds play entails respecting the child’s behavioral state and following the child’s lead by attending to the child’s vocalizations, prompts, and other behavior cues. Infants engage in coordinated adult–child interactions from birth, and by 3 months of age they already have expectations of mutuality with reciprocal receptions of and reactions to cues during turn taking [25,26]. Adults can facilitate and increase a child’s participation in interactions by creating structured, rhythmic turn taking sequences that are well synchronized with the child’s responses [27,28]. Within this synchronization, it is important that the adult provides pauses to allow the child to process information, respond or make choices, or even take a break, if needed [28,29]. Salient, unambiguous prompts, and allowing the child ample response time expands the child’s opportunities for exploration and mutuality during interaction [25,30]. This is especially important when working with children with motor delays or multiple disabilities since high frequencies of prompts can be overwhelming [29,30]. These children may need increased time for information processing due to deficits across steps of attention, recognition, recall, encoding, integration, and/or motor planning [30,31,32,33].

Children with motor delays may also present with expressive impairments that limit their ability to communicate during play and make it more challenging for the adult to pick up on the child’s intentional acts and signs of engagement or distress [34]. Synchrony in interaction requires the adult to learn about and understand the child’s less obvious communicative signals (i.e., eye movements, breathing patterns, gestures, vocalizations, and protesting behaviors), then support the child’s relaying of these messages and appropriately interpret whether these signals indicate child engagement, dis-engagement, or distress during play [35].

Tactile stimulation and touch are integral within adult–child play interactions. For children with motor delays, tactile stimuli both increase and disturb a child’s attention during social play [36]. Given the individual variations in sensory processing among children with motor delays, it is important to determine the appropriate amount and type of tactile stimulation beneficial to the individual child, and to elucidate when such stimuli become overwhelming, leading to dis-engagement or distress [36]. Provenzi et al. [36] classified maternal touch into categories including: affective, playful, facilitating, and holding. Among these, playful touch such as tickling, squeezing, moving, or flexing the child’s body was associated with increased attentiveness during social play [36]. Such touch might also increase a child’s attention during PT that embeds play. A study of therapeutic touch in pediatric PT shows that flexible, subtle handling during play can awaken the child’s curiosity and facilitate new motor explorations [37].

#### 3.1.2. Respecting the Child’s Autonomous Play Initiatives and Engagements

Children develop autonomous exploratory behavior as part of their ongoing spontaneous play which is guided by perceptual input, motor output, and the consequences of actions that the child attends to [38]. Infants as young as 2 months of age demonstrate exploratory play engagement by gazing at activities only when they result in interesting consequences [38]. By 7 to 9 months of age, infants engage with more solitary object exploration [8,25], and by 10 months, infants tend to be more responsive and engaged in joint attention during free play compared to semi-structured play [39]. When adults provide directions during play, simple and structured directions best maintain the children’s exploratory behavior, as opposed to unstructured or more complex directions [40]. In a study of 3- to 14-month-old infants born preterm, Håkstad et al. [11] noted that to uphold the child’s play engagement during therapy sessions, physical therapists need to engage in mutual play and coordinate their actions with the child’s play intentions and goals. These findings underscore the importance of allowing children to discover and direct play and to decide the extent of time spent with a play activity, without intrusiveness from the adult play partner.

#### 3.1.3. Including Activities across Developmental Domains

PT that embeds play should include activities that facilitate development across domains. Facilitation of perceptual-motor exploratory behavior, or motor-based problem-solving, along with socioemotional support to help children self-regulate and manage frustrations, is a priority [29]. Play interactions can provide a substrate for rich language and social environments. As adults narrate play activities (e.g., naming objects, describing actions, or counting objects), children learn about shared attention and turn taking, begin to understand that objects have names and actions, and that their body’s actions interact with the world to make things happen [29,41]. The use of motor skills such as sitting and reaching, or self-initiated mobility during play, create developmental cascades in language, social, visual-perceptual, and/or cognitive skills as the child engages with the environment or others.

#### 3.1.4. Supporting a Child’s Engagement in Play

Mirroring and supporting a child’s use of toys assists in engaging the child in play. Mimicry, vocal cues, and pointing support attention maintenance, joint attention, and joint interaction [29,42,43]. Mimicry allows the formation of a social connection and facilitates future interactions. Eighteen-month-old children are more likely to invite an adult when the adult has previously mimicked the child’s use of a toy [42]. Mimicry activates the mirror neuron system as the child observes the adult play partner imitate the child’s actions. This supports language, social, and emotional development, and observational learning [44]. Vocal cues also support sustained infant attention during play with objects [43] and joint attention in toddlers [45]. Deak and colleagues [45] found that 15- and 21-month-olds responded more to parental gaze-shifting with pointing or with directed language than gaze-shifting alone, highlighting the importance of vocal cues and gesturing during play. In early infancy prior to gaining locomotion, infants are reliant on adults to present toys and objects for exploration; adults organize the infant’s interaction. These interactions are often rhythmic, providing structure and facilitating the infant’s play with both the object and the adult play partner [28].

Creating the “just right challenge” during play is essential for child engagement during PT that embeds play. If an activity is too challenging, either from a motor or cognitive perspective, the therapist risks losing the child’s interest. Conversely, an activity that is too easy may not provide the therapeutic effects intended by the therapist [46]. The activity should engage and motivate the child, and the child should be able to master the skill with their “focused effort” [47]. Physical therapists can scaffold the “just right challenge” by grading motor or cognitive aspects of the task to meet the child’s abilities and adapting the activity to the appropriate level for the child [48,49]. The START-Play intervention incorporated the “just right challenge” to scaffold blended motor/cognitive skills, and to engage families in brainstorming about how to increase the difficulty of activities through small, achievable increments. These small increments support advances in motor and cognitive skills. The position of the child during play is an important consideration when creating a “just right challenge”. Increased motor demand may reduce opportunity for social interaction and reduce the child’s cognitive capacity for motor-based problem solving [50].

Attraction to novel stimuli is an adaptive behavior that intrinsically motivates a child to explore their environment [51,52]. Infants demonstrate a desire to explore in the first weeks of life [53] and a preference for novel stimuli, habituating to what is regular or expected and paying particular attention to what is unusual [51,54,55]. Using novel stimuli to elicit exploration requires ensuring an appropriate familiarization time with the previous stimulus [56]. Rose and colleagues [56] found that infants demonstrate a preference for familiar stimuli when a shortened familiarization period is given. Increased familiarization time is required for younger infants due to slower processing speeds and when introducing complex stimuli [51]. This concept is pertinent for therapists practicing PT that embeds play. Inadvertently switching from one novel activity to another without appropriate familiarization time for the child may lead to a child’s preference to return to a previous task and disengage from the new task. This is especially key when working with children who have known cognitive impairments, as they may require more time to explore and process a new object or task due to decreased processing speed [51,54].

#### 3.1.5. Adapting Play to Individual Child Differences

PT that embeds play requires clinicians to consider individual differences in play based on the child’s cognitive and sensory-motor abilities. Physical therapists may need to identify alternative stimuli to initiate play, incorporate adaptive toys, use external supportive equipment, and systematically alter activities to find the “just right challenge” for each child. Alternative stimuli may be necessary for children who have sensory deficits such as visual impairments. For example, Hughes [57] encouraged clinicians to create a “sensory-rich play environment” for children with visual impairments. This includes incorporating sensory cues to guide exploration such as changes in the texture of flooring to provide tactile input to a mobile infant with visual impairments. Clinicians also are encouraged to consider the tactile and/or auditory properties of a toy rather than the appearance alone when choosing toys. Allowing a child with sensory impairment to safely explore their environment rather than deterring them is important to ensure continual development of intrinsically motivated locomotion. Finally, beginning with one play partner, perhaps a familiar adult, prior to increasing the number of play participants is important to not overwhelm a child with sensory impairments [57]. A PT who embeds play can support and facilitate play while allowing the parent or caregiver to be the play partner for the child to avoid overwhelming the child and risking subsequent child disengagement from the activity.

Using a strength-based approach is important to motivate the child and family to participate in play. A clinician who engages in a strength-based approach focuses on the child’s strengths and assets, rather than their deficits, and incorporates play activities accordingly [58]. Not only does the strength-based focus allow clinicians to identify areas to facilitate development, it also can increase parental well-being and positive interactions between the parent and child. Steiner and colleagues [59] found that using a strength-based approach with families of children with autism spectrum disorder improved parent affect and parent–child interactions, with significant findings for increased physical affection and positive affect as compared to those who received a deficit-based approach.

### 3.2. Focus, Environment, and Toy Use

#### 3.2.1. The Focus of Play

In PT that embeds play, the child’s movement is a means to engage in play, not the primary focus of the child’s attention. When focusing primarily on movement patterns or repetitions, physical therapists may inadvertently interrupt or limit a child’s play. This inclination away from play and exploration towards movement repetitions becomes frustrating for children who are able to recognize that an adult is intentionally withholding a toy [60]. In contrast, children learn better when adults recognize and respond to the child’s communicative gestures and allow the child to be actively engaged in acquiring information that is salient to them [61]. Allowing the child to select how to play within a PT session may remove control from the physical therapist but provides autonomy to the child, leading to decreased frustration and improved learning.

#### 3.2.2. The Environmental Set-Up

A key role of the physical therapist is to set up the environment and materials in a way that allows the child to initiate play and then to explain to parents and caregivers about why and how we do this. Similarly to setting up the “just right challenge” for any task, physical therapists must consider the “just right environment” [62]. It is important to consider the physical and psychosocial environment to ensure the child has the opportunity to explore and engage in complex interactions [53]. Additionally, the therapist or caregiver can guide and enable the child’s play behavior within this environment [62]. One factor that affects a child’s ability to interact with the environment is body position. A child in prone plays differently than a child in sitting or in supine due to the constraints of the position [53]. A child who has balance difficulties in standing may support themselves with two hands and not engage with toys or may discover the possibility of leaning on the support surface to free their arms to engage [63]. A physical therapist can assess how the child’s position impacts their ability to play and interact with the environment to determine if increased support is needed or if a different position is warranted.

The position of the therapist or caregiver and how much support they provide a child also has an impact on the child’s ability to explore [63].. For example, children in supported sitting, particularly those who are not well supported, do not touch or reach for objects in their environment as much as independent sitters [64]. Modulating support to match the child’s motor and cognitive needs is an important aspect of play within a PT context. Additionally, face-to-face interactions are important so that the caregiver can read the child’s behavioral and visual cues and vice versa [65]. In many supported play positions, the caregiver is behind the child, leading to decreased joint attention to an object and to each other’s cues, and to a decreased ability for the caregiver to facilitate and scaffold the child’s play behavior [66]. The use of an external support may allow a caregiver to move to a position that allows for eye contact and increased quality and scaffolding of play.

#### 3.2.3. The Use of Toys

Play can occur with or without toys. Social games such as Peak-a-boo are play opportunities that have a clear recurring structure that young infants recognize [67] and enjoy as long as the established routine is followed [68]. Physical therapists can use social games to establish a playful atmosphere even with very young infants. Play can also occur as a child actively explores and interacts with toys and objects, discovering and exploring movement possibilities, environmental opportunities, and their own autonomy [69,70,71]. Physical therapists can interrupt or limit a child’s play by using toys as bait to encourage movement repetitions without allowing the child to fully explore the toy before it is moved again. This not only removes autonomy from the child, but also leads to frustration and missed opportunities to learn through exploration. Therapists may help extend the play with the toy that the child is interested in and support movement when the child is ready to transition to a new activity. As few as four to five toys [72] may be appropriate to support movement as larger numbers of toys may create less focus and less creative play [72,73].

### 3.3. Engaging Families in Play

#### 3.3.1. Understanding a Family’s Play Culture

Play is deeply enculturated. Who plays, how they play, why they play and what they play with is influenced by societal, geographical, sociodemographic, familial, and individual belief systems and values [74]. Adult eye contact, expression of emotion, use of narratives, and physical proximity or touch during play differs across cultures and across families, parents, and children based on child temperament, gender, and/or birth order [75,76,77]. Play in Western cultures largely is defined as a child-initiated, child-led learning activity focusing on self-discovery, object exploration, and/or social interaction [70,75,78]. Families of higher socioeconomic status link play to resources: safe, physical indoor and outdoor play spaces, a wide variety of play materials, and caregivers who have the time for and understanding of developmental play [79]. Families in low-income settings are less likely to have access to these resources. Perceived gender or social roles, of both the parent and the child, interact with the players’ individual temperaments and beliefs, shaping overt play behaviors [80]. The reasons, or why of play, and the materials used, or what is played with, are similarly related to cultural factors [74,79]. Some families do not value play and view it as frivolous; while others value it as an opportunity to scaffold learning and development [78,81,82]. Some prioritize physical activity; others prioritize cognitive, social-emotional or fine motor skills as predictors of success at school age [75,76]. What is played with can vary widely based on custom, resources, or preference. Sicart [83] argues that anything (a toy, a household item, another person) may become a ‘plaything’ or object of relational interaction. From such perspective, it is not what is played with but how the child interacts with the ‘plaything’ that matters.

Finally, parents’ perceptions of child well-being or capabilities influence all the above aspects of play. Children with multiple medical conditions or diagnosed with motor delays are more likely to be perceived by their families as vulnerable [80]. Parents with this perception are less likely to introduce play behaviors that involve risk and are less likely to choose play items or motor activities that appropriately challenge their children [84,85]. They are also more likely to underestimate their child’s physical, developmental or play abilities and to interrupt or control play activities [84,85]. If physical therapists fail to probe for, acknowledge, or attend family play culture, they risk designing intervention or play programs of little practical value.

#### 3.3.2. Partnering with Families

Partnering is a multi-faceted, family-centered process that promotes family engagement in intervention programs [86,87]. It implies ‘co-construction’ of the therapeutic relationship: shared observation of child and family strengths/needs, shared development of therapy goals and outcomes, and shared conceptualization of the role of intervention in the child’s development [88,89]. It acknowledges that the primary agent for a child’s developmental change is the parent–child relationship [86,90,91]. A parent’s contingent responsiveness, or ability to read and respond to their child’s cues, is related to both secure relationships and the degree to which very young children explore their environments [91,92,93]. Responsive parents extend play and promote early learning through attentive but non-directive interactions [93]. These carefully nuanced interactions enhance the child’s mastery motivation, tolerance to frustration, and focused attention: all skills associated with stronger cognitive, communication, and self-regulatory/adaptive developmental outcomes [92,93]. Partnering between professionals and parents implies transparency, equality in decision making, and absolute ‘presence’ [87].

Presence may be considered physical proximity. Anecdotally, therapists and parents often comment that parental presence in the PT session is distracting to the young child. However, motor learning suggests that behaviors observed during therapist–child interactions are capacity-related and not true performance [94]. There is no guarantee of carry-over into daily routines if parents are not actively involved. Additionally, children rely on familiar caregivers to understand the context of any social interaction. When separated from their parents, they are not as likely to read the subtle shifts in gaze or to respond to the tactile cues that familiar caregivers use to direct, attenuate, and shift attention [92,93,95] needed for both social-emotional regulation and learning [93,96].

Presence may be considered attentional. In the therapeutic relationship, attentional demands are complex, dynamic and triadic (parent-child-therapist) [88,91]. Parents’ and therapists’ attention at any given moment is potentially fragmented by many things: other responsibilities, worries about the immediate and distant future, and constant technoference, defined as cell or smart-phone disruptions during social interactions [97]. In the parent–child relationship, these interruptions can lead to increased child distress, disrupted infant social-emotional regulation, lowered child inhibitory responses, and impaired contingency-related learning of both language and social cues [97,98,99]. Simply put, attentional disruptions interfere with the ability to recognize and respond to a child’s cues during play or any therapeutic interaction.

Engaged parents extend the reach and dose of any intervention, including play by embedding therapeutic activities into daily routines [89,90]. Parents who participate in play-based intervention programs report gaining an understanding of quality play time, spending more time with their child during play activities, and having a greater understanding of the developmental impact of play [100]. Multiple frameworks for engaging families exist. King et al. [87,88] propose four key principles for family engagement: (1) the personalizing principle, or ‘knowing the client’; (2) the individual variation principle, or knowing that clients differ in how they demonstrate engagement and what engages them; (3) the relationship principle, or that engagement is cultivated through interpersonal relationships; and (4) the monitoring principle, or staying attuned to the child’s and the parent’s level of engagement from moment-to-moment and from session-to-session. Practical strategies for family engagement as described by Marvin et al. include open communication, encouraging parent–child interaction during sessions, overtly linking play behaviors to developmental or motor outcomes and modeling, suggesting and practicing play behaviors in action and together. Therapists can invite parents to play, describe the purpose and learning opportunities embedded in play, and affirm parents in their parent–child play interactions. For children with motor impairments, this may include teaching parents to ‘wait’, to allow their child opportunities for trial and error and to support their child’s focused attention during play.

## 4. Conclusions

PT that embeds play has the potential to support acquisition of skills across the developmental continuum. In this framework, therapists work to engage the child in play, to facilitate optimal environmental set-up and toy selection, and to engage the family in play interactions with their child. Consideration of these components ensures the therapist is supporting play and development and is not disrupting or interfering with the play of children.

Currently, a gap exists in our understanding of PT that embeds play. Research suggests there may be differences in how physical therapists interpret and implement play in the design of PT sessions with children [11,15]. Additional information on if and how physical therapists intentionally incorporate play within their assessment and intervention sessions should be examined in greater depth. Educational guidelines for how physical therapists should be prepared to incorporate and support the play of children with or at high risk of motor delays are lacking, which may contribute to variation and a disconnect between effective strategies and clinical practice trends.

Future research should explore perceptions of both clinicians and of parents related to PT that embeds play. Greater understanding of therapist beliefs related to the importance of play in facilitating developmental skills, their own playfulness during therapy, and their comfort and skill in supporting the parent’s ability to play with their child is needed. Additionally, the perceptions of parents related to play and how best to facilitate their interactions with their children should be explored. Together, this information may inform additional guidelines or research to inform how best to support optimal play and overall development of infants and toddlers with or at risk of motor delays.

## Data Availability

Not applicable.

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
