# Peer review of "Embedding Play to Enrich Physical Therapy"

_behavsci, 2023, doi:10.3390/bs13060440_

Round 1

Reviewer 1 Report

Thank you very much for inviting me to review this article. Which I consider to be very complete and can greatly help pediatric physiotherapists around the world to understand the importance of play in the development of children; having to improve their clinical practices to search for the implementation of the game together with the neurophysiological processes of learning in the natural environment of the child and where parents have a fundamental role.

I would like to contribute a series of aspects that could improve the article:

- Currently there are tools such as AHEMD or AHEMD-IS that allow us to know the natural environment of the child together with the toys used by the parents, to establish a relationship with motor development. So it may be interesting to mention these tools when assessing the environment and how we can influence it.

- It would be interesting to mention the neurophysiological principles of imitation as a game, that is, to talk about the mirror neuron complex.

- Currently, there are certain projects like GAME and SPEEDI that include many of the principles mentioned in their study, so please consider mentioning them.

- It can be very interesting to talk about certain references that I leave below since it could contribute neurophysiological aspects of early age and play in ASD

References

- Early intervention evidence for infants with or at risk for cerebral palsy: an overview of systematic reviews

- Effectivity of Play‑Based Interventions in Children with Autism Spectrum Disorder and Their Parents A Systematic Review

Regards

Author Response

Currently there are tools such as AHEMD or AHEMD-IS that allow us to know the natural environment of the child together with the toys used by parents, to establish a relationship with motor development. So it may be interesting to mention these tools when assessing the environment and how we can influence it. Author response: Thank you for this suggestion. We are familiar with the assessment tools mentioned. However, we have opted not to include any information related to specific assessment tools as it would be impossible to comment on all viable tools within the scope of this manuscript. 

It would be interesting to mention neurophysiological principles of imitation as a game, that is, to talk about the mirror neuron complex. Author response: The following statement was added to: 3.1.4 Supporting a Child’s Engagement in Play: Mimicry activates the mirror neuron system as the child observes the adult play partner imitate the child’s actions. This supports language, social, and emotional development and observational learning.

Currently, there are certain projects like GAME and SPEEDI that include many of the principles mentioned in their study, so please consider mentioning them. Author response: Thank you for this suggestion. We have added the following statement in the text under, 2.1 Play in Pediatric Physical Therapy: Recent research on interventions to support early development including, Supporting Play Exploration and Early Developmental Intervention (SPEEDI)[13] and Goal Activity Motor Enrichment (GAME)[14] studies, incorporate principles of play as important components of the intervention, yet define play differently, highlighting the importance and need for additional consideration of play.

It can be very interesting to talk about certain references that I leave below since it could contribute neurophysiological aspects of early age and play in ASD. References: - Early intervention evidence for infants with or at risk for cerebral palsy: an overview of systematic reviews. - Effectivity of play-based interventions in children with autism spectrum disorder and their parents: a systematic review. Author response: We appreciate these suggestions, and while we agree that these recent review papers examine interventions that include play, the focus of this paper is broadly on motor delay. We have attempted to limit the discussion throughout the paper to intervention concepts rather than providing an overview specific to each diagnosis. We have opted not to add these two reviews for this reason.  

Reviewer 2 Report

I am very pleased to have the opportunity to review this manuscript. This manuscript, entitled "Embedding Play to Enrich Physical Therapy," describes the importance of play in pediatric physical therapy.

I have read the manuscript and would like to comment as follows

1.(Summary)

Where and by whom was the 3-day consensus meeting held? If this would affect the character limit in the abstract section, please indicate in the text.

2.(Definition)

Children are defined as under 3 years of age. Please define what is considered pediatric physical therapy in the field of physical therapy.

3.(2.1 Play in Pediatric Physical Therapy)

As a related matter to what is described in this section, you may want to consider this based on general pediatric physical therapy session times.

Author Response

  1. (Summary). Where and by whom was the 3-day consensus meeting held? If this would affect the character limit in the abstract section, please indicate in the text. Author response: A statement indicating the consensus meeting was organized and hosted by the Motor Development Lab at the University of Southern California was added to the text in the Introduction. 
  2. (Definition). Children are defined as under 3 years of age. Please define what is considered pediatric physical therapy in the field of physical therapy. Author response: In section 2.1 Play in Pediatric Physical Therapy, a phrase was added to define pediatric physical therapy as traditionally serving children and youth birth to 21-years of age. 
  3. (2.1 Play in Pediatric Physical Therapy). As a related matter to what is described in this section, you may want to consider this based on general pediatric physical therapy session times. Author response: Thank you for your comments, however, we are unclear of the meaning of this request. If vital, please provide additional guidance.